# In Vitro, Ex Vivo, and In Vivo Models for the Study of Pemphigus

**DOI:** 10.3390/ijms23137044

**Published:** 2022-06-24

**Authors:** Roberta Lotti, Claudio Giacinto Atene, Emma Dorotea Zanfi, Matteo Bertesi, Tommaso Zanocco-Marani

**Affiliations:** 1DermoLAB, Department of Surgical, Medical, Dental and Morphological Sciences, University of Modena and Reggio Emilia, 41124 Modena, Italy; 2Hematology Section, Department of Medical and Surgical Sciences, University of Modena and Reggio Emilia, 41124 Modena, Italy; claatene@unimore.it; 3Department of Life Sciences, University of Modena and Reggio Emilia, 41125 Modena, Italy; emma.zanfi@gmail.com (E.D.Z.); matteo.bertesi@unimore.it (M.B.); zanocco@unimore.it (T.Z.-M.)

**Keywords:** pemphigus vulgaris, animal model, autoimmune disease, autoantibodies, atypical pemphigus, mucocutaneous pemphigus, desmoglein, desmocollin

## Abstract

Pemphigus is a life-threatening autoimmune disease. Several phenotypic variants are part of this family of bullous disorders. The disease is mainly mediated by pathogenic autoantibodies, but is also directed against two desmosomal adhesion proteins, desmoglein 1 (DSG1) and 3 (DSG3), which are expressed in the skin and mucosae. By binding to their antigens, autoantibodies induce the separation of keratinocytes, in a process known as acantholysis. The two main Pemphigus variants are Pemphigus vulgaris and foliaceus. Several models of Pemphigus have been described: in vitro, ex vivo and in vivo, passive or active mouse models. Although no model is ideal, different models display specific characteristics that are useful for testing different hypotheses regarding the initiation of Pemphigus, or to evaluate the efficacy of experimental therapies. Different disease models also allow us to evaluate the pathogenicity of specific Pemphigus autoantibodies, or to investigate the role of previously not described autoantigens. The aim of this review is to provide an overview of Pemphigus disease models, with the main focus being on active models and their potential to reproduce different disease subgroups, based on the involvement of different autoantigens.

## 1. Introduction

Pemphigus is a severe autoimmune disease. Among the different phenotypic variants encompassing this disease, Pemphigus vulgaris (PV) and Pemphigus foliaceus (PF), caused by the presence of autoantibodies against the desmosomal cadherins desmoglein 3 (DSG3) and 1 (DSG1), are the most common forms [1,2,3]. PV and PF are IgG dependent, while other forms of Pemphigus can be IgA dependent. This group of diseases targets epithelia. Stratified epithelia and mucous membranes are particularly affected. The pathology typically causes acantholysis, a process where keratinocytes lose cell–cell adherence, thereby causing the detachment of specific layers of the epithelia, triggering the formation of blisters. Disruption of the skin and/or mucosae through blistering varies in the different forms of Pemphigus, depending on the cadherins affected. For instance, in the presence of anti DSG1 antibodies, blistering predominantly occurs in the skin (PF). If DSG3 is targeted, blistering mainly involves mucosae (mucosal PV), while if antibodies against both DSG3 and DSG1 are present, both skin and mucosae are affected (mucocutaneous PV). Cadherins take part in the formation of desmosomes, the structures of which allow cell–cell contact in the epithelia. Desmosomes do not only provide mechanical stability to the epithelia, but are also involved in cell signalling pathways. Consequently, Pemphigus autoantibodies do not trigger blistering just by inhibiting cadherin interaction, thereby destabilizing desmosomes, but also by altering intracellular signalling; desmosomes, in fact, are involved in multiple signalling pathways, such as Ca⁺⁺/PKC, apoptosis signalling, as well as modulations of p38 MAPK, EGFR, and heat shock protein 27, thus mediating the proliferation and differentiation of keratinocytes [4,5,6]. Although DSG3 and DSG1 are the cadherins mainly targeted by autoantibodies, other cadherins, such as Desmocollin1 (DSC1), DSC2 and DSC3, are involved in nonclassical forms of Pemphigus [7]. Beyond cadherins, other proteins, such as anti-cholinergic receptors, have been described as targets for autoantibodies in Pemphigus patients [8]. The role of autoantibodies against non-cadherin autoantigens in the disease has not been completely clarified yet, but their presence further highlights the complexity of this disease.

Pemphigus may lead to death if untreated. The first-line therapy is the use of systemic corticosteroids (CS) alone or in combination with immunosuppressant adjuvants. CS impinge upon multiple transduction pathways, producing immunosuppressive, anti-inflammatory, anti-proliferative and vasoconstrictive effects, which can lead to severe adverse effects. Rituximab, the chimeric monoclonal anti-CD20 antibody, is another drug used in the treatment of Pemphigus, at different dosages, as an adjuvant with CS therapy. However, this is not exempt from adverse effects [9]. Other treatments include plasmapheresis, used to remove pathogenetic IgGs from plasma or immunoadsorption, by removing IgGs from blood via its passage in adsorption columns containing molecules with high affinity to IgGs [10]. Another non-immunosuppressive approach is the intravenous administration of immunoglobulins, derived from plasma rich in IgG and poor in IgA and IgM. However, this approach has drawbacks too [11]. Recently, the neonatal Fc receptor (FcRn), which plays a role in IgG homeostasis, preventing the lysosomal degradation of IgGs, thereby prolonging their circulating life and regulating their serum levels, has been considered as a target to develop new therapies for autoimmune diseases [12,13,14]. Moreover, it was recently observed that, in epidermal keratinocytes in PV, its function would not be based solely on autoantibody recycling, but also on more direct effects on cell adhesion [15]. In this regard, FcRn antagonists have been developed and represent an interesting, new experimental possibility in the therapy of Pemphigus. Nevertheless, effective treatments stabilizing desmosomes, or limiting autoantibody activities, are still greatly needed.

Pemphigus is a very complex and heterogeneous disease, where not only several autoantigens and autoantibodies are involved, but frequently a mixture of IgG antibodies in various antigens is present in patients. Over the course of the disease, and due to different treatments, the antigen profile and the pathogenicity may substantially change. Therefore, to study the role of autoantibodies found in PV (PVIgG), to address the function of different autoantigens in the genesis of the disease, and to test the efficacy and mode of action of experimental therapies, several experimental models have been developed. These can be in vitro, ex vivo or in vivo, passive, or active. The aim of this review is to provide an overview of Pemphigus disease models, with a focus on active models and their potential to reproduce different disease subgroups, based on the involvement of different autoantigens.

## 2. In Vitro and Ex Vivo Assays for the Evaluation of the Pathogenicity of Pemphigus Antibodies and the Efficacy of Experimental Drugs

For many years, it has been well known that cultures of human skin treated with Pemphigus antibodies undergo acantholysis. This technique demonstrated that, for example, hydrocortisone present in a skin culture was able to inhibit acantholysis induced by Pemphigus sera [16]. This approach was further ameliorated and standardized, leading to the establishment of a protocol allowing acantholysis to be induced in human skin organ cultures (HSOC) injected with a bi-specific anti-DSG 1 and three single-chain antibody variable fragments (scFv) [17], PVIgG or the pathogenic IgG AK23 [18] (Figure 1A). Using this method, blister formation can be easily evaluated by immunofluorescence or haematoxylin and eosin staining. One of the main advantages of the HSOC method is that the architecture of human skin is preserved. Moreover, by injecting the pathogenetic antibody into the skin, rather than letting it diffuse into the medium of culture, it is possible to reach all the cells almost simultaneously. The HSOC protocol allows the evaluation of the therapeutic potential of experimental drugs, inhibiting blister formation. This can be achieved by comparing the extent of the epidermal split of the positive control sample to that of the epidermis treated with the test article.

Other in vitro protocols to evaluate the pathogenicity of antibodies, or the activity of drugs, are the induction of DSG3 internalization and the keratinocyte dissociation assay, both performed using either normal human epidermal keratinocytes (NHEK) or HaCaT cells (Figure 1B,C). The DSG internalization assay [19] was inspired by an observation suggesting that binding of the autoantibodies leads to internalization of DSG3 and collapse of the keratin cytoskeleton [20]. By this assay, keratinocytes are cultured at specific Ca2+ concentrations to induce the formation of cell contacts and differentiation of keratinocytes. Cells are then exposed to PVIgG. Successive internalization of DSG3 is evaluated by immunofluorescence, where only cells exposing DSG3 on the surface will be positive (Figure 1B). By this assay, it is possible to test the activity of drugs hypothetically inhibiting the formation of blisters. Test articles would be administered to the cell culture prior to the treatment with PVIgG.

The keratinocyte dissociation assay or dispase-based keratinocyte dissociation assay (DDA) was originally developed to study cell–cell interactions [21], and was successfully adapted to the study of Pemphigus [22]. The idea behind this protocol is to evaluate the effect of different stimuli on the loss of keratinocyte adhesion in cell monolayers. DDA requires the generation of a cell monolayer; administration of different stimuli, such as PVIgG, scFv or AK23; dispase-based detachment of cell monolayers from the cell culture dish; application of mechanical stress by standardized pipetting for monolayer fragmentation; and quantification of fragments (Figure 1C). For these methods, both NHEK and cell lines such as HaCaT can be used. Primary keratinocytes might represent the primary choice since they provide variability that better reflects the in vivo situation, although cell lines have an advantage in terms of cost, ease of handling and reproducibility. DDA is a commonly used assay whose major flaw is inter- and intra-experimental comparability. Very recently, a standardized protocol has been proposed to perform DDA under optimal conditions [23].

Considering the different characteristics of the in vitro/ex vivo approaches described so far, the best option, when testing the efficacy of putative therapies, is to compare test articles with different methods. This was recently performed to screen the efficacy of inhibiting the blister-inducing activity of autoantibodies of low-molecular-weight molecules, isolated from a library of 141 substances inhibiting different signalling pathways [24]. The efficacy of such substances was filtered through the DSG3 internalization assay and DDA performed in HaCaT cells and NHEK, allowing the identification of four novel therapeutic targets for the modulation of PVIgG-induced intraepidermal blistering. Among the four, one was validated by the HSOC method.

Recently, in a further development of the organ cultures method, HSOC and human oral mucosa organ cultures injected with PVIgGs deriving from patients with mucocutaneous PV, mucosal-dominant PV or AK23, treated or untreated with p38 MAPK inhibitors, demonstrated that in the human epidermis, blistering can be prevented by the inhibition of p38MAPK, while in the mucosa, PVIgG and AK23 induce blisters via a mechanism not dependent on p38MAPK, highlighting, once again, the versatility of these methods [25,26].

## 3. In Vivo Passive and Active Models for the Study of Pemphigus

As an alternative to the in vitro models discussed previously, in vivo models provide the possibility to observe the progression of the disease and to test putative drugs inside a living organism over time. The first in vivo models of Pemphigus were based on evidence that neonatal mice injected with Pemphigus IgG develop skin blisters [27,28] (Figure 2A). This kind of model is considered “passive”, since PVIgG antibodies are transferred from a donor with the disease to the animal and are not directly produced by mice themselves, in contrast to “active models”, where mice are able to produce autoantibodies. The neonatal model is fast and easy to make, since in around twenty hours, new-born, wild-type mice develop Pemphigus lesions.

DSG3^null^ mice have a phenotype superimposable to PV patients [29], as regards lesions to the mucous membrane, but cannot be considered a Pemphigus model since the pathogenetic mechanism is completely different to that of Pemphigus. Another approach consists of injecting PVIgG into full-thickness human skin xenografted onto severe combined immunodeficient (SCID) mice [30,31]. This approach was not further validated, due to the complexity of the xenograft model, and to the fact that it does not add much to a classical passive model.

Passive models have been widely used to investigate pathogenetic mechanisms and signalling of Pemphigus. For instance, passive transfer of PVIgG to normal and DSG3^null^ neonatal mice was used to demonstrate that if both DSG1 and DSG3 are expressed concomitantly, antibodies against either alone are not sufficient to determine spontaneous blistering [32].

Since neonatal mice have not finalized epidermal morphogenesis, the transfer of PVIgG into neonatal mice does not allow for the study of lesions to mature hair follicles and stem cell niches. To deal with these questions, a model was developed with passive transfer of AK23 into adult 8-week-old animals [33]. Passive transfer experiments were also performed in transgenic mice expressing humanized DSG3 (hDSG3). In this model, human hDSG3 transgenic animals are crossed with the murine DSG3 knockout line. It was observed that mucosal PV (mPV) sera bind mucosal epithelia from the hDSG3 mice, but not mucosal tissues from WT mice, confirming the pathogenicity of mPV anti-hDSG3 IgG in vivo [34]. Humanized mice were also generated using Pemphigus-related human class II HLAs, which led to the observation that if mice were humanized with an unrelated human HLA, they did not produce specific antibodies against DSG3 after immunization. Thus, it is possible to state that recognition of DSG3 protein epitopes by CD4+ cells is related to the HLA haplotype [35,36]. In vitro assays, such as the dispase-based dissociation assay, and the in vivo passive transfer model into neonatal mice were used jointly to verify the capacity of DSG1- and DSG3-specific adsorbers to remove circulating pathogenic autoantibodies from three PV patients [10].

Since passive transfer models can only be studied for a limited time, active models where animals are able to directly produce autoantibodies have been developed and are more suitable for the study of experimental drugs over a prolonged period. To generate an active model for PV, a DSG3^null^ mouse can be immunized against the knock-out protein via the injection of murine recombinant DSG3 (rDSG3), to obtain auto-reactive antibodies through in vivo production [37] (Figure 2C). Mice are immunized with repeated doses of recombinant protein, combined with different adjuvants in various immunization steps. After this first process, the immunized animals are sacrificed and the splenocytes are collected and transferred into immunodeficient Rag2^−/−^ mice through tail vein injection or, in some cases, intraperitoneally; these splenocytes will continue to produce antibodies against the protein used for immunization in the immunodeficient mouse, which expresses DSG3 [38]. It has also been demonstrated that, by this method, anti-DSG3 antibodies with different pathogenicity are produced, and that pathogenic heterogeneity among anti-DSG3 IgG is due to their epitopes [39].

In addition to the active model obtained by immunization of knock-out (KO) animals with a recombinant protein, it was also observed that the adoptive transfer of naive splenocytes from non-immunized DSG3^null^ mice induces anti-DSG3 IgG production and the PV phenotype in RAG2^−/−^ mice [40,41] (Figure 2B). Therefore, DSG3-specific naive lymphocytes in DSG3^null^ mice can be primed and activated by the endogenous DSG3 in recipient mice to produce pathogenic anti-DSG3 IgG without active immunization. This study evidenced that the active immunization step with recombinant DSG3 is probably beneficial to increase the numbers of primed DSG3-specific T and B cells, and that the primed DSG3-specific T and B cells are more efficiently stimulated in recipient mice than naive cells. However, it suggests that the active immunization step is not essential when enough naive splenocytes are transferred, making this a simpler and less time-consuming alternative. Interestingly, the pathogenic activity in blister formation of different anti-DSG3 IgG antibodies obtained from PV model mice generated by the transfer of naïve DSG3^null^ splenocytes was evaluated in the passive transfer model using neonatal mice [42].

The active DSG3 mouse model was successfully used in several studies. It helped to demonstrate that loss of tolerance against DSG3 in both B and T cells is crucial for the development of pathogenic anti-DSG3 IgG [43], and that a single potent DSG3-reactive T cell is sufficient to commit polyclonal naïve B cells to produce pathogenic anti-DSG3 IgG and induce PV phenotype in mice [44]. Using the DSG3 active model, it was shown that anti-DSG3 antibodies can directly access DSG3 present in desmosomes in vivo, and cause the desmosome separation that leads to the formation of blisters [38]. The active DSG3 model was also used as a tool to evaluate various immunosuppressive therapies, commonly used for the treatment of patients [45]. The active DSG3 model was also used to investigate the efficacy of intravenous Ig (IVIG), a treatment option for untreatable cases of PV [46], showing that IVIG decreases circulating anti-DSG3 titre after 5 days of treatment.

Pemphigus is a complex and heterogeneous disease. Beyond PV and PF, which are the most common forms, characterized by the presence of autoantibodies against DSG3 and 1, there are forms of atypical Pemphigus where patients develop autoantibodies against other autoantigens, such as the cadherin desmocollin (DSC), cholinergic receptors, mitochondrial proteins, and others [47]. Whether these atypical autoantibodies are responsible for specific forms of the disease, or whether they are associated with the disease and participate, to some extent, in the pathological phenotype, remains to be clarified. In this regard, active models might prove to be useful thanks to their plasticity. For example, Lotti et al. recently described a version of the active model allowing the production of autoantibodies against desmocollin 3 (DSC3). In this case, to obtain splenocytes reactive against DSC3, WT mice were injected with rDSC3 to break the tolerance to DSC3 to produce pathological antibodies [48] (Figure 2D). After collection, splenocytes were injected into Rag2^−/−^ mice. This method enabled the adoptive transfer of autoreactive DSC3 lymphocytes to Rag2^−/−^ immunodeficient mice, that normally express DSC3. The presence of circulating anti-DSC3 auto-antibodies is sufficient to determine the appearance of a pathological phenotype relatable to Pemphigus, but with features not completely superimposable to those observed in the DSG3 active model, suggesting that the DSC3 active model might mimic the atypical Pemphigus. Moreover, when a mixture of splenocytes derived from DSC3-immunized animals and DSG3-immunized mice were injected in the same Rag2^−/−^ mouse, the presence of both anti-DSC3 and anti-DSG3 antibodies determines a more severe phenotype and a slower response to prednisolone. Extending this approach to other antigens will indeed provide information on the role played by atypical autoantibodies in Pemphigus.

## 4. Conclusions

Over the last few years, both in vivo and in vitro models for the study of Pemphigus (summarized in Table 1) have been further developed and perfectioned. Assays such as DSG internalization and dispase-based keratinocyte dissociation (DDA), which are less costly and time consuming than in vivo models, have been highly dissected, and protocols allowing satisfactory reproducibility have been described, based both on the cell type chosen and on experimental conditions. In addition to these assays, human skin organ cultures (HSOC) allow blistering to be observed in the complete layers of skin, rather than in a monolayer of cells. These approaches proved their usefulness, especially in preliminary studies of therapies preventing the formation of blisters.

In vivo models are far more demanding. The main flaw of the passive transfer of IgG is the inability to follow up the animals for a long period of time, and the inability of the animals to produce their own autoantibodies. These flaws are overcome by active models, where animals are engrafted with splenocytes, allowing the production of autoantibodies against selected autoantigens. Of course, these models are time consuming, and require experience and a number of animals per experimental group sufficient to provide statistical significance to the data obtained. So far, as active models are concerned, the DSG3 PV model, the DSC3 atypical Pemphigus model, and the DSG3/DSC3 model have been described. Among the issues to be addressed when trying to compare different active models is the intensity of the phenotype. For instance, the DSG3 model is obtained by injecting into Rag2^−/−^ splenocytes deriving from DSG3^−/−^ mice immunized against rDSG3, while the DSC3 model is obtained by injecting into Rag2^−/−^ mice splenocytes deriving from WT mice that underwent breakage of DSC3 tolerance. It should be taken into account that these different immunization strategies might lead to different levels of IgG production, depending on partial loss of tolerance obtained by tolerance breakage, compared to introducing a neo-antigen in antigen null animals. In addition, the number of splenocytes transferred also matters and should be taken into consideration when planning an experiment.

Active models have a high potential thanks to their plasticity; there are also reports in the literature that question the central role of DSG3 in PV [49]. Consequently, it would be even more interesting to broaden active models to other autoantigens, in order to further clarify their role. Active models could also represent a good template to be expanded to other autoimmune diseases. Moreover, besides being a good testing ground for experimental therapies, they also could prove to be useful to study the mechanisms underlying Pemphigus, for instance, not just the role of B cells, but also that played by T cells that are also represented among the splenocytes transferred during the establishment of the model.

## Figures and Tables

**Figure 1 ijms-23-07044-f001:**
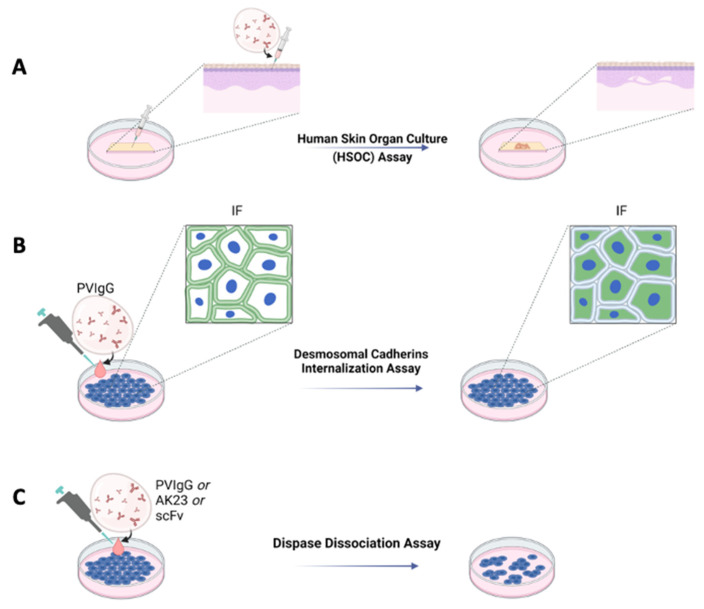
Ex vivo and in vitro schematic representations of Pemphigus models. (**A**) Ex vivo model with human skin organ culture (HSOC) assay. IgG fractions prepared from patients’ sera (PVIgG), or other engineered anti-DSG3 and/or DSG1 antibodies (i.e., scFv), are injected subcutaneously into human healthy skin. Blisters showing the typical histology observed in patients appear after 24 h. (**B**) In vitro desmosomal cadherins internalization assay in normal human keratinocyte or HaCaT culture monolayers. (**C**) In vitro with dispase dissociation assay. IgG fractions prepared from patients’ sera (PVIgG), or monoclonal antibodies directed against desmoglein 3 (DSG3) (i.e., AK23) or other engineered anti-desmoglein 3 (DSG3) and/or DSG1 antibodies (scFv), are used in normal human keratinocyte or HaCaT culture monolayers.

**Figure 2 ijms-23-07044-f002:**
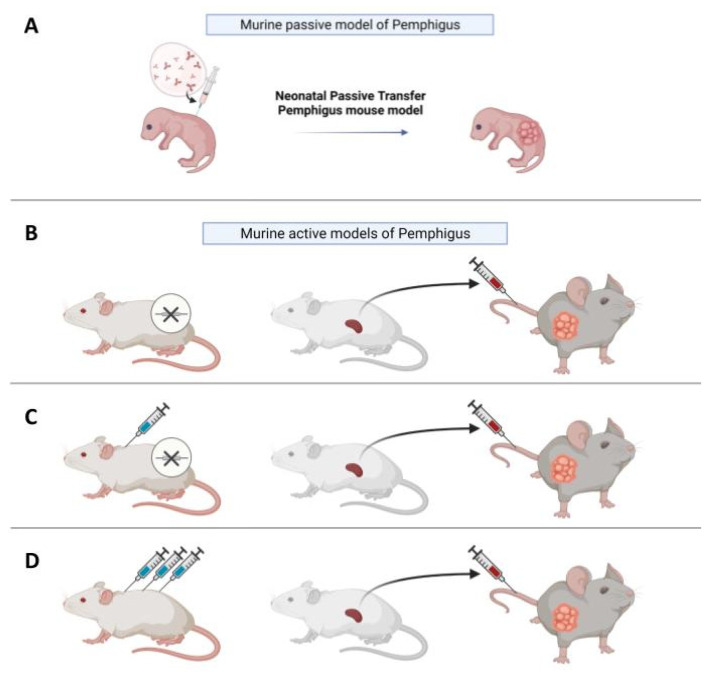
In vivo models of Pemphigus. (**A**) In vivo neonatal passive transfer Pemphigus mouse model. IgG fractions isolated from patients′ sera (PVIgG), or other engineered anti-desmoglein 3 (DSG3) antibodies, are injected subcutaneously or intraperitoneally into new-born mice. Mice develop blisters in 24–36 h, which show the typical histology observed in patients. (**B**) Active disease model. DSG3^null^ mice are not tolerant against DSG3 given that DSG3 is never exposed to the immune system. After adoptive transfer of naive lymphocytes from DSG3^null^ mice, Rag2^−/−^ immunodeficient mice produce anti-DSG3 IgG antibodies and display traits of PV phenotype. (**C**) Another active disease model is generated by adoptive transfer of splenocytes after repeated immunization with recombinant DSG3 of a DSG3^null^ mouse. The receiving Rag2^-/-^ mouse will develop blisters and hair loss because, in mice, intercellular adhesion of follicular epidermis is mainly mediated by DSG3 during the rest phase of hair growth. (**D**) Active model after tolerance breakage in a WT mouse with high doses of antigen. After repeated injections, splenocytes were transferred in Rag2^−/−^ mice, which would develop blisters.

**Table 1 ijms-23-07044-t001:** Schematic summary of the models currently available to study Pemphigus.

Model	In Vitro Ex Vivo	In Vivo	Cell Culture or Mice	IgG Production	Treatment	Advantages	Disadvantages	Application
Human skin cultures	X		Healthy cells	No	Pemphigus antibodies from patients	Acantholysis study	Limited time	Therapies preventing acantholysis
HSOC	X		Healthy cells in organ cultures	No	scFv, PVIgG, AK23	Skin structure preserved, acantholysis and blistering study	Limited time, complex	Therapies inhibiting blistering
DSG internalization assay	X		NHEK or HaCaT	No	PVIgG	Internalization of DSG3	Limited time	Pathogenesis, Therapies inhibiting blistering
DDA	X		NHEK or HaCaT	No	PVIgG, scFv or AK23	Keratinocyte dissociation studies	Limited time, reproducibility	Pathogenesis, Therapies inhibiting blistering
Passive		X	Neonatal mice	No	IgG from patients	Development of pemphigus lesions	Does not allow to study lesions in mature hair follicles and stem cell niche, limited time	Pathogenesis and Signalling of Pemphigus
			SCID	No	PVIgG	Development of pemphigus lesions	Not further validated, limited time	
			Adult 8-week-old	No	AK23	Lesions in mature hair follicles and stem cell niche	Limited time, expensive	
			Human hDsg3 transgenic	No	Mucosal PV sera	Mucosal PV sera bind mucosal epithelia from the hDSG3 mice	Limited time, expensive	
Active for PV		X	DSG3^null^ & Rag2^−/−^	Against DSG3	Splenocytes from DSG3^null^ immunized mice into Rag2^−/−^	Production of IgG against different epitopes. Longer follow up	Complex and time consuming	Pathogenesis and signalling of pemphigus. Study of experimental drugs over a prolonged period.
			DSG3^null^ & Rag2^−/−^	Against DSG3	Naive splenocytes from non-immunized DSG3^null^ mice into Rag2^−/−^	No immunization. Long Follow up	Lack of immunization makes a less efficient model	
Active for Atypical Pemphigus		X	WT & Rag2^−/−^	Against DSC3	Tolerance break against DSC3 in WT mice. Splenocytes transferred into Rag2^−/−^	Production of IgG against different epitopes. Longer follow up	Complex and time consuming	
			WT, DSG3^null^ & Rag2^−/−^	Against DSC3 & DSG3	Splenocytes from both WT and DSG3^null^ immunized mice transferred into Rag2^−/−^	Production of IgG against different epitopes. Longer follow up	Complex and time consuming	

## Data Availability

Not applicable.

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
