# Peer review of "In Vitro, Ex Vivo, and In Vivo Models for the Study of Pemphigus"

_ijms, 2022, doi:10.3390/ijms23137044_

Round 1

Reviewer 1 Report

The manuscript “In Vitro, Ex Vivo, and In Vivo Models for the Study of Pemphigus” submitted by Roberta Lotti et al sets out to review the literature to provide an overview of Pemphigus disease models.

A great effort has been made to analyze the literature, even the very recent one, and one quickly realizes the quality of the work done.

The figures (and table) are very nice and important, and give the reader a quick overview of what can be expected using this or that method.

The literature analyzed is rich and abundant and the authors have well detailed the method of analysis and selection of publications.

I only have one small comment, which is that the authors start naming Figure 1C before Figure 1B or 1A in the text. I think you should change the arrangement of your figure 1 to and move Human skin organ culture to position 1A.

I thank the authors for this review.

Author Response

We changed the order of models in figure 1. Legends and text have been changed accordingly.

Reviewer 2 Report

Line 33: The paraneoplastic pemphigus is predominantly IgG, not IgA, dependent.

Line 39: It is still controversial whether blistering in the different forms of pemphigus depends on the cadherins affected.

Line 57: If untreated, pemphigus leads to death.

Line 213: The phenotypes in RAG2-/- mice primed with anti-DSG3 autoimmunity are not consistent with human pemphigus. The authors can find details about deficiencies of animal models of DSG3 autoimmunity in the review paper titled "Pseudo pemphigus phenotypes in mice with inactivated desmoglein 3: further insight to the complexity of pemphigus pathophysiology" published in Am J Pathol (PMID 26506474).

Author Response

Line 33: The paraneoplastic pemphigus is predominantly IgG, not IgA, dependent. We corrected accordingly

Line 39: It is still controversial whether blistering in the different forms of pemphigus depends on the cadherins affected. We agree, in fact in the text we were cautious and we feel like we didn't use dogmatic expressions

Line 57: If untreated, pemphigus leads to death. we amended accordingly

Line 213: The phenotypes in RAG2-/- mice primed with anti-DSG3 autoimmunity are not consistent with human pemphigus. The authors can find details about deficiencies of animal models of DSG3 autoimmunity in the review paper titled "Pseudo pemphigus phenotypes in mice with inactivated desmoglein 3: further insight to the complexity of pemphigus pathophysiology" published in Am J Pathol (PMID 26506474). We briefly discussed this topic in the conclusion and added the reference.